# Lidar Cloud Detection with Fully Convolutional Networks

**Erol Cromwell & Donna Flynn**
Pacific Northwest National Laboratory
Richland, WA 99354, USA
{erol.cromwell,donna.flynn}@pnnl.gov

## Abstract

In this contribution, we present a novel approach for segmenting laser radar (lidar) imagery into geometric time-height cloud locations with a fully convolutional network (FCN). We describe a semi-supervised learning method to train the FCN by: pre-training the classification layers of the FCN with "weakly labeled" lidar data, using "unsupervised" pre-training with the cloud locations of the Wang & Sassen (2001) cloud mask algorithm, and fully supervised learning with hand-labeled cloud locations. We show the model achieves higher levels of cloud identification compared to the cloud mask algorithm.

## 1 Introduction

The vertical distribution of clouds from active remote sensing instrumentation is a widely used data product from global atmospheric measuring sites. The presence of clouds can be expressed as a binary cloud mask and is a primary input for climate modeling efforts and cloud formation studies. Current cloud detection algorithms producing these masks do not accurately identify the cloud boundaries and tend to oversample or over-represent the cloud. Additionally, they require significant effort to develop and maintain and are sensitive to instrument changes and accurate instrument calibration. Machine learning has recently been applied to cloud detection, but in a limited setting with supervised learning (Gómez-Chova et al., 2017). However, it has not been applied to cloud detection from ground-based lidar instruments.

In our work, we present a novel method for using FCNs to detect clouds from lidar data that surpasses the well-established Wang & Sassen (2001) cloud mask algorithm. FCNs have been shown to accurately segment images semantically with pixel-to-pixel predictions (Long et al., 2015) and we use a similar approach to segment clouds from lidar imagery. We develop a semi-supervised learning method to train the FCN, involving pre-training the "classification" weights of the model and pre-training the entire model with "unsupervised" learning. Weakly-supervised and semi-supervised learning techniques have been successfully applied to deep convolutional networks for image segmentation (Hong et al., 2015; Papandreou et al., 2015). Likewise, unsupervised pre-training of FCNs has been shown to improve model performance (Wiehman et al., 2016). Both approaches allow model training that leverages large amounts of unlabeled data while requiring minimal ground truth data. This is important since manually create a binary cloud mask for lidar data is a time and labor intensive process, which results in limited ground truth data available.

## 2 Experiment

**Dataset**  We use micropulse lidar (MPL) data from the 30smplcmask1zwang datastream from the Southern Great Plains (SGP) C1 facility (January 2010 to December 2016) to train the model (Sivaraman & Comstock, 2011a). One 30smplcmask1zwang data file contains a 24-hour period of lidar profiles at 30 second temporal resolution and 30m vertical resolution out to 18km (667 x 2880 data points). The total attenuated backscatter and linear depolarization ratio (LDR) from the datastream are the input data. We also use the Wang & Sassen (2001) algorithm cloud mask (Sivaraman & Comstock, 2011b) from the datastream for comparison and in the training process. A lidar interpreter expert hand-labeled cloud mask for 85 days to use as ground truth to train and

test the model. 55 days (January-February 2015) are used for training and validation and 29 (March 2015) are held out for testing. We increase the amount of hand-labeled data eightfold by splitting each day into quarters with some overlap (667 x 800) and adding the horizontally flipped version to the dataset.

**Model architecture**    We base our model architecture on the U-net model introduced by Ronneberger et al. (2015). Figure 1 details our model design. The model input is the quarter day backscatter profile and LDR data (667 x 800 x 2). The output is the softmax probability distribution for each time-height point. The distribution for the two classes of whether a point is: (a) a cloud, or (b) not a cloud. A point is classified as a cloud if the probability of being in the cloud class is greater than or equal to 50%. We modified the U-net model to accommodate our input and output size by adding a convolutional layer reducing the data size to 640 x 800 x 2 and a deconvolution layer to the end of the model returning the data to its original dimensions.

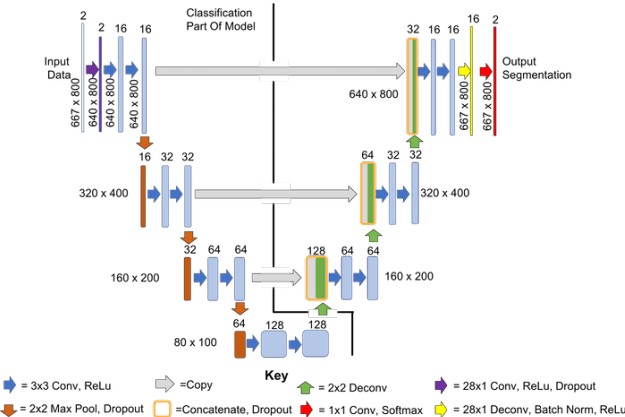

Figure 1: Diagram of FCN trained to identify clouds from lidar data

**Training**    The FCN model is trained in three stages. First, the classification part of the model identified in figure 1 is pre-trained using a set of 1780 "weakly-labeled" quarter days (890 containing cloud, 890 without cloud). A quarter day contains clouds if the algorithm cloud mask contains clouds in the time period. Second, we perform "unsupervised" learning on the FCN segmentation model with a set of 4200 quarter days (4000 with clouds, 200 without clouds), using the algorithm cloud mask as our ground truth. This is "unsupervised" because the algorithm cloud mask is already calculated and and reproducing the algorithm is not the end goal of the model. Finally, the FCN segmentation model is trained and fine-tuned using 432 quarter days (362 with cloud, 70 without) from January and February 2015, using the hand-labeled cloud mask as ground truth. The loss function for training the classification weights is the prediction accuracy and for the entire FCN model is categorical cross entropy.

**Metrics**    For the FCN segmentation model, we use precision (% of predicted clouds that are actual clouds), recall (% of clouds that are predicted as clouds), and f1-score (harmonic mean of the precision and recall) as our performance metrics. Each of the metrics are calculated over the entire dataset, and not per quarter day. We use F1-score instead of accuracy for the FCN segmentation model because only a small percentage (approx. %5) of the time-height points in the hand-labeled cloud mask are clouds, which skews the accuracy unreasonably high.

## 3    RESULTS & DISCUSSION

Overall, the FCN segmentation model outperforms the Wang & Sassen (2001) algorithm cloud mask. For the holdout dataset, the FCN model has an F1-score of 0.8508, exceeding the performance of the algorithm (0.65) as seen in row 1 and 4 of table 1, respectively. The model precision is almost double, indicating the model correctly identifies more clouds than the algorithm at twice the rate. Thus, the model captures more of the cloud detail in the output than the algorithm cloud mask. We note that the model slightly underperforms against the algorithm in recall (0.8687 and 0.9049,

Table 1: Model perfomance on holdout dataset (March 2015)

| Method | F1-Score | Precision | Recall |
|---|---|---|---|
| Wang & Sassen (2001) algorithm | 0.65 | 0.5072 | 0.9049 |
| No pre-training | 0.8263 | 0.7801 | 0.8783 |
| Without "unsupervised" learning | 0.8242 | 0.7938 | 0.857 |
| FCN segmentation model | 0.8508 | 0.8336 | 0.8687 |

Figure 2: Cloud segmentation results for several days (top to bottom: March 13th, March 16th, March 26th, 2015). (a): MPL backscatter profile. (b): MPL linear depolarizartion ratio. (c): hand-labeled cloud mask. (d): segmentation model output. (e): algorithm cloud mask.

respectively). However, the algorithm's higher recall is at the expense of exaggerating the cloud tops and bottoms and merging multiple cloud layers. Figure 2 presents several qualitative results from the model. As shown, the clouds identified by the model closely follow that of the hand-labeled mask. In the first and third examples (March 13th and March 26th), the model output is more detailed in contrast to the algorithm cloud mask product, which tends to exaggerate the cloud shape and size. In the second example (March 16th), the algorithm is unable to consistently detect the cloud layer as indicated by the vertical gaps in the mask.

To verify the training methodology, we also trained a FCN model without the first two training steps (i.e., no pre-training) and one without the "unsupervised" learning step. As seen in table 1 (rows 2 and 3), both of these models have lower F1-scores (0.8263 and 0.8242) and precision (0.7801 and 0.7938) than the fully trained FCN model. While the FCN model with no pre-training does have a slightly higher recall (0.8783) than the fully trained model, it is more important for the model to correctly predict clouds than to identify all of the clouds. Thus, the classification pre-training and the "unsupervised" pre-training increase the model's overall performance.

## 4 CONCLUSION

We successfully used FCNs to segment clouds from lidar imagery. We showed our novel semi-supervised training method outperformed the Wang & Sassen (2001) cloud mask algorithm and improved the overall performance of the model. We have begun initial work analyzing how the model performs on lidar data from different observations sites (i.e. mid-latitude vs. polar) and potential transfer learning to make the model more robust. Initial results are promising, but require further investigation. Additionally, we want to investigate if data seasonality (winter vs. summer) impacts model training and cloud detection.

ACKNOWLEDGMENTS

The research described in this abstract is part of the Deep Science for Scientific Discovery Initiative at Pacific Northwest National Laboratory. It was conducted under the Laboratory Directed Research and Development Program at PNNL, a multiprogram national laboratory operated by Battelle for the U.S. Department of Energy. We would like to thank the Pacific Northwest National Laboratory 2017 Quickstarter Inititave for initial funding for this work. Additionally, we would like to thank the Atmospheric Radiation Measurement (ARM) Data Center for the 30smplcmask1zwang datastream data. This research was performed using PNNL Institutional Computing at Pacific Northwest National Laboratory. and the model was implemented with the Keras python module (Chollet et al., 2015).

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
