# OpenReview forum: "Lidar Cloud Detection with Fully Convolutional Networks"
_ICLR.cc/2018/Workshop — Reject_

### Official Review · AnonReviewer3 · 2018-03-08
**Not a deep learning contribution**

**Rating:** 4
**Confidence:** 4

**Review:**

This may be a nice contribution in the domain of cloud location estimation, but it's not a deep learning contribution as it merely used an existing deep network. Hence I don't think it fits the ICLR workshop goal.

---

### Official Review · AnonReviewer2 · 2018-03-09
**Makes use of unlabeled data to improve performane, but not in an especially novel way**

**Rating:** 6
**Confidence:** 3

**Review:**

The paper describes a method for segmentation of cloud locations in LIDAR images using a fully convolutional network. The network itself is based on the well-known U-net architecture. The main novelty in the paper is the use of auxiliary tasks to pretrain the network, which acts as a regularizer and results in improved performance. The first of these auxiliary tasks is to train the encoder part of the network simply to classify whether there are cloud locations present. The second is to train on noisy labels generated by an hand engineered segmentation algorithm. The paper demonstrates that this approach outperforms the hand engineered solution by a wide margin. Ablative studies illustrate the importance of the auxiliary tasks.

Strengths: approach is semi-supervised and makes use of unlabeled and weakly labelled data; outperforms the baseline by a wide margin.

Weaknesses: the architecture is not novel and the use of auxiliary tasks and weak labels is not new; some details are missing (see comments below).

Comments:
- I presume the classification part of the network has a dense layer attached with a sigmoid unit to predict binary labels and these are compared with the weak labels using cross entropy, but this is not made explicit.
- The paper is missing some details about the training procedure (number of epochs, learning rates, learning algorithm, momentum, etc.)
- I assume that the March test data was not used for the "unsupervised" and weakly supervised training phase, but this wasn't made clear in the paper.
- There's no need to use a softmax and categorical cross entropy when there are only two classes. Just use a sigmoid and binary cross entropy. Doing this reduces the amount of redundant parameters.

Note: I am not familiar with LIDAR segmentation so cannot comment on the performance or novelty wrt the state of the art.

---

### Official Review · AnonReviewer1 · 2018-03-09
**A new problem domain for ML researchers.**

**Rating:** 6
**Confidence:** 2

**Review:**

This paper proposes an FCN to solve the could detection problem. The main contribution of this paper is the new problem domain and dataset which are not widely known to ML researchers. However, the proposed approach is not very novel in terms of the modeling aspect.
I listed some comments below:
- The second step is not a generally accepted notion of "unsupervised" pre-training. In general, when we pre-train the model we would expect more like a reconstruction of input from output, however, the proposed model uses some sort of noisy labels which are obtained by "algorithm cloud mask" (which is again not very clear in the context although I understood it as the algorithm proposed by Wang and Sassen). To reduce the confusion from the notation, it would be better to reformulate the first and second step as pre-training with 'image-level annotation' and 'noisy annotation', respectively.
- As a workshop paper which introduces a new application domain, it would be more interesting if the authors include more details about the problem domain and datasets, since most of the workshop participants may not be familiar with the given problem, and why the FCN has to be used. For example, the dataset seems unique in the sense that the sensed image does not contain a horizontal aspect of sky image and only provide a time-height view of the sky. So one can understand the data as a sequence of vectors which might be more appropriate to be modeled by RNN instead of CNN.

---

### Decision · Program_Chairs · 2018-03-20
**ICLR 2018 Workshop Acceptance Decision**

**Decision:**

Reject

**Comment:**

Based on the reviews, this paper has not been accepted for presentation at the ICLR workshop. However, the conversation and updates can continue to appear here on OpenReview.